# High Expression of IL-1RI and EP_2_ Receptors in the IL-1β/COX-2 Pathway, and a New Alternative to Non-Steroidal Drugs—Osthole in Inhibition COX-2

**DOI:** 10.3390/ijms20010186

**Published:** 2019-01-07

**Authors:** Natalia Karolina Kordulewska, Anna Cieślińska, Ewa Fiedorowicz, Beata Jarmołowska, Elżbieta Kostyra

**Affiliations:** Department of Biology and Biotechnology, University of Warmia and Mazury, Oczapowskiego 1A Street, 10-719 Olsztyn, Poland; natalia.smulska@uwm.edu.pl (N.K.K.); anna.cieslinska@uwm.edu.pl (A.C.); flowerpower20@o2.pl (E.F.); bj58@wp.pl (B.J.)

**Keywords:** PBMCs, histamine, anti-inflammation, cultures in vitro, expression

## Abstract

Background: Osthole (7-methoxy-8-isopentenylcoumarin) is natural coumarin isolated from the fruit of *Cnidium monnieri* (L.) *Cusson*, which is commonly used in medical practice of traditional Chinese medicine (TCM) in various diseases including allergies and asthma disorders. Purpose: Osthole was tested for the anti-histamine, anti-allergic, and inhibitory effects of COX-2 (cyclooxygenase-2) in children with diagnosed allergies. Additionally, we hypothesize that stated alterations in children with diagnosed allergies including increased expression of interleukin 1-β receptor type 1 (*IL-1 type I*) and *E-prostanoid* (*EP*) *2* receptors, as well as raised expression, production, and activity of COX-2 and IL-1β in incubated medium are approximately connected. Furthermore, we establish the mechanisms included in the changed regulation of the COX-2 pathway and determine whether osthole may be COX-2 inhibitor in peripheral blood mononuclear cells (PBMCs). Method: PBMCs were obtained from peripheral blood of healthy children (control, n = 28) and patients with diagnosed allergies (allergy, n = 30). Expression of the autocrine loop components regulating PGE_2_ production and signaling namely *IL-1 type I receptor* (*IL-1RI*), *cyclooksygenaze-2* (*COX-2*), *E-prostanoid* (*EP*) *2*, and also *histamine receptor-1* (*HRH-1*) was assessed at baseline and after stimulation with histamine, osthole, and a mixture of histamine/osthole 1:2 (*v/v*). This comprised the expression of histamine receptor 1 (*HRH-1),*
*IL-1RI, COX-2, EP*_2_ receptor, and the secretion of IL-1β and COX-2 in cultured media and sera. Results: Compared with control group, basal mRNA expression levels of *HRH-1, IL-1RI, COX-2*, and *EP*_2_ were higher in the allergy group. Histamine-induced *EP*_2_ and *COX-2* expression mRNA levels were also increased. Conclusions: Osthole successively inhibits *PGE*_2_ and *COX-2* mRNA expression. Furthermore, osthole reduces the secretion of COX-2 protein in signaling cellular mechanisms. Changed *EP*_2_ expression in children with allergies provides higher *IL-1RI* induction, increasing IL-1β capacity to increase COX-2 expression. This effects in higher PGE_2_ production, which in turn increases its capability to induce IL-1RI.

## 1. Introduction

The global prevalence of allergic diseases has increased recently and in such frequency that current explanations based on genetic changes do not provide sufficient understanding of this phenomenon [1]. The causes of this increase remain unclear and although the “*hygiene hypothesis*” has received significant attention, this does not provide appropriate immunological explanation for the observed rise in T helper type 2 (Th2)-polarized disease [2]. While the pathogenic mechanism underlying this disorder is not fully understood, it appears closely related to imbalance in the arachidonic acid (AA) metabolism, and more specifically, prostanoid and cysteinyl leukotriene metabolism [3].

Prostaglandin (PG) E_2_ is a metabolite generated from AA through the action of COX enzymes and PGE synthesis. Under inflammatory conditions, there is an accompanying increase in COX-2 expression, which ensures enhanced PGE_2_ production. The biological activities of PGE_2_ are mediated by four receptors: E-prostanoid (EP_1_), EP_2_, EP_3_, and EP_4_ [4]. 

IL-1β is a proinflammatory cytokine produced and secreted by a variety of cell types [5,6]. It is produced as the pro–IL-1β inactive precursor and is activated via proinflammatory protease caspase-1 to active IL-1β. Mature IL-1β leaves the cell and activates target cells through interactions with two membrane-bound receptors: IL-1 receptor type I (IL-1RI) and IL-1 receptor accessory protein [7]. These events are responsible for the expression of a wide range of genes involved in inflammation, and previous studies have shown that IL-1β is the primary cytokine involved in COX-2 upregulation in inflammatory processes [8,9,10] and that it induces COX-2 expression [11]. 

Interestingly, various studies have demonstrated strong interdependence between IL-1β, IL-1RI, and COX-2 regulation. The expression of IL-1RI is modulated by IL-1 and PGE_2_ [12], and COX-2 and mPGES-1 expression and PGE_2_ production and IL-1RI expression levels appear directly related. Moreover, COX-2 expression is in a positive feedback loop with PGE_2_ which increases the COX-2 expression through its EP_2_ and EP_4_ receptors. These activate adenylate cyclase and consequently increase cellular cyclic AMP (cAMP) concentration, and this increases COX-2 mRNA levels by activating COX-2 transcription and increasing COX-2 mRNA stability [13].

Our search for anti-allergic agents from natural sources revealed promising properties in the *Cnidium monnieri* dried fruit and the isolated substance named osthole, which has an isopentenoxy-coumarin structure. Pharmacological studies demonstrate its wide bioactivity as an anti-osteoporotic, anti-carcinogenic, anti-diabetic, and anti-allergic agent [14,15,16,17]. We hypothesize osthole has potential in allergy treatment in inhibition in COX-2 pathway. 

We hypothesize that alterations in the expression of components in the COX pathway are related events in children with diagnosed allergies. Based on the central role of the EP_2_ receptor in the regulation of the COX-2 autocrine positive feedback loop, we also consider that abnormal expression of the EP_2_ receptor is responsible for the altered regulation of the COX pathway.

## 2. Results

### 2.1. Basal Expression of HRH-1, IL-1RI, COX-2, and EP_2_ Receptors

After three days of incubation, we detected that *HRH-1*, *IL-1RI*, COX-2, and *EP_2_* receptors showed significantly higher expression in the allergy group compared to control (*p* < 0.0001) (Figure 1).

### 2.2. HRH-1 Gene Expression Induced by Histamine

PBMC cells were incubated with and without histamine (150 ng/mL), osthole (300 ng/mL), and histamine/osthole 1:2 (*v/v*) for 72 h to analyze their effect on *HRH-1* mRNA expression and this was measured using real-time PCR (Figure 2A). In the control group, histamine displayed a 2.6-fold increased expression of *HRH-1* mRNA compared to cells without stimulation. In the allergy group, histamine increased 2.8-fold. We did not observe significant differences between the level of *HRH-1* mRNA expression after histamine stimulation between the control and allergy groups. Osthole effect

Expression of *HRH-1* was significantly lower after stimulation with osthole compared to PBMCs cultured with histamine in the control and allergy groups. We also observed a greater effect of osthole than histamine in the mixture of those two compounds (Figure 2A).

### 2.3. IL-1RI Gene Expression

Induced by histamine

Cultured PBMCs were incubated with and without histamine (150 ng/mL), osthole (300 ng/mL), and histamine/osthole 1:2 (*v/v*) for 72 h to analyze their effect on *IL-1RI* mRNA expression (Figure 2B). Incubation of control group PBMCs with histamine significantly increased *IL-1RI* expression 6.5-fold compared to cells without stimulation. In the allergy group, histamine had no effect on PBMCs, and this result can indicate abnormal expression of *IL-1RI* in the allergy group. Osthole effect

Expression of *IL-1RI* was significantly lower after stimulation with osthole compared to PBMCs cultured with histamine in the control group. We observed a greater effect of osthole than histamine (Figure 2B).

### 2.4. COX-2 Gene Expression

Induced by histamine

As described by Kordulewska (Data not shown) [18], our results showed increased induction of the *COX-2* gene expression in response to histamine in children with diagnosed ASD with co-existing allergies. The same result was observed in the allergy group, where histamine showed a 3.34-fold increased expression of *COX-2* mRNA in PBMCs. Moreover, quantitative real-time PCR analysis of histamine-induced *COX-2* mRNA expression revealed COX-2 levels significantly lower in the control group than in children with allergies (Figure 2C). Osthole effect 

Incubation of the allergy group’s PBMCs with 300 ng/mL osthole significantly decreased *COX-2* mRNA gene expression compared to those incubated with 150 ng/mL histamine. The histamine/osthole mixture also produced this decrease; again, highlighting the inhibitory effect of osthole on histamine in cultured cells, though we did not report significant differences in control group (Figure 2C).

### 2.5. EP_2_ Gene Expression 

Induced by histamine

Incubation with 150 ng/mL histamine significantly increased *EP*_2_ gene expression 4-fold in the allergy group cultured PBMCs that were not stimulated (Figure 2D). Osthole effect

Osthole significantly decreased *EP_2_* gene expression in the allergy group compared to cells treated with histamine. The result was also noted in PBMCs incubated with the histamine/osthole mixture. This emphasized the greater effect of osthole than histamine. In addition, *EP*_2_ gene expression in the control group was unaffected by stimulation with the tested substances, even with histamine exerting no effect on the allergy group PBMCs (Figure 2D).

### 2.6. IL-1β Concentration

In medium

We observed a significant 16.3-fold increase in IL-1β concentration in the allergy group compared to the control group, and this was similar to PBMCs behavior after stimulation with the tested substances (Figure 3A). While histamine significantly induced IL-1β concentration in the allergy group compared to PBMCs incubated in pure medium, osthole significantly decreased the amount of interleukin. In serum

A *p* < 0.0001 significant difference was recorded between the control and allergy groups in IL-1B serum concentration (Figure 3B). 

### 2.7. COX-2 Concentration

In medium

Significant increases in COX-2 concentration were noted in allergy group PBMC’s cultured with pure medium, histamine, and osthole compared to the control group (Figure 4A). While histamine significantly induced COX-2 concentration in both allergy and control groups compared to PBMC’s incubated in pure medium, osthole significantly decreased this concentration in both groups compared to PBMC’s incubated with histamine. We also determined that mixed histamine/osthole PBMC culture in the allergy group exerted the same effect as cells incubated with osthole alone. In serum

Significant differences were noted in serum COX-2 between control and allergy groups (*p* < 0.0001) (Figure 4B).

### 2.8. The Inhibitory Effects of COX-2—Activity of COX-2

To confirm that osthole is able to inhibit COX-2 in cultured cells, we checked the obtained results from substances we analyzed with reference compound DuP-679. DuP-679 is a member of the diaryl heterocycle group of selective COX-2 inhibitors.

In medium RPMI, we found that COX-2 was significantly inhibited by osthole and the reference compound DuP-679. Furthermore, we did not observe statistically significant differences between the inhibition of DuP-679 or osthole (Figure 5A). These findings indicated that there may be the same model of action for osthole like DuP-673 on the COX-2 inhibitory effect, which supports our earlier hypotheses about the antiallergic and anti-inflammatory effect of osthole.

Again, in cultured PBMC cells, we observed that osthole statistically inhibited COX-2 in both analysed groups compared to the pure enzyme (Figure 5B).

## 3. Discussion

Although the mechanisms responsible for allergies are not fully understood, induced PGE_2_ production offers some explanation [17,19,20] and our group hypothesize that anomalies in increased mRNA gene *EP_2_* and *IL-1RI* expression are also involved because all IL-1β activity currently identified appears mediated through its only known functional receptor, IL-1RI [7,9]. Further, because of the importance of this cytokine in COX-2 upregulation in increasing PGE_2_ production, we examined relationships between *IL-1RI**, COX-2*, and *EP*_2_ mRNA gene expression and IL-1β and the COX-2 concentration in medium/serum. To confirm that osthole can modulate the COX-2 pathway, given what was already found in mice [21], we examined the inhibitory effect of COX-2 using reference compound DuP-679.

First of all, we demonstrated that the basal expression levels of *HRH-1*, *IL-1RI*, *COX-2*, and *EP_2_* mRNA were lower in the control compared to the allergy group. The overexpression of IL-1RI mRNA could contribute to the induction of COX-2 and EP_2_ mRNA expression. Furthermore, after cellular stimulation with histamine, we observed an increase in *HRH-1* mRNA levels in both the control and allergy groups, and COX-2 and EP_2_ only in the allergy group. However, IL-1RI mRNA showed higher levels after histamine stimulation only in the control group.

Considering the capacity of histamine to modulate *HRH-1* mRNA expression, we also investigated whether osthole could minimalize the histamine effect on HRH-1 receptor. We observed that osthole significantly reduced cellular stimulation and effects on HRH-1. This dependence can be used in future in designing new anti-allergic drugs, which mainly create an influence by blocking the HRH-1 receptor. 

We demonstrated that PBMCs isolated from the allergy group after histamine stimulation had reduced levels of *IL-1RI* mRNA expression compared to the allergy group. This deficient *IL-1RI* expression, seen only in the allergy group, most likely contributed to higher basal expression of *IL-1RI* gene expression. This finding is in accordance with previous studies because literature data asserts that histamine contributes to the progression of allergic–inflammatory responses by enhancing secretion of pro-inflammatory cytokines, such as IL-1α, IL-1β, and IL-6, and also chemokines including RANTES and IL-8 [22]. Additionally, it is likely that body fluid histamine levels in allergy-diagnosed children are so frequent and high that IL-1RI receptors do not react. A further possibility is that the higher IL-1β level in these children is connected with deficient IL-1RI receptor induction. Our control group results agree with previous studies performed on the histamine/IL-1RI receptor expression positive feedback system. The osthole and histamine/osthole mixture successively reduced *IL-1RI* gene expression in the control group compared to PBMCs incubated with histamine alone, so we hypothesize that osthole suppressed receptor expression. 

Although there is no documented analysis of *COX-2* mRNA expression in allergy-afflicted children and controls following stimulation by the tested substances, COX-2 plays a key role in inflammation. Dinarello [23] records that IL-1β is a potent pro-inflammatory cytokine crucial for host defense responses to infection, allergic inflammation and injury, and Martin and Wesche [24] report IL-1β induced COX-2 expression is a well-described phenomenon resulting from translocation of transcription factor-κB to the nucleus.

We also tested histamine pro-inflammatory mediation in our research model. Quantitative real-time PCR revealed that histamine significantly increased *COX-2* mRNA expression in the allergy group. This suggests that the histamine response is dysfunctional in allergy-afflicted children, and this is supported in literature assertions where histamine activates allergic inflammation and increases *COX-2* gene expression in allergic children. In contrast, osthole and combined osthole/histamine reduces COX-2 expression, and thus osthole can be effective in the treatment of allergic disease by decreasing PGE_2_ levels. Supporting findings include: (1) Ariasnegrete et al. [25] observed this relationship, where macrophages had 8-fold increased *COX-2* expression after stimulation by lipopolysaccharides. Their additional finding of decreased *COX-2* mRNA expression in dexamethasone stimulated cells is comparable to our results where osthole decreased PBMC *COX*-2 expression compared to histamine. Dexamethasone is recognized as an effective corticosteroid treatment for rheumatic abnormalities, some skin diseases, asthma, and severe allergies. (2) Similar results came from Payvandi et al.’s. [26] study, where PBMC cells stimulated by LPS, IL-1β, and TNF-α had increased *COX-2* expression. (3) Kordulewska et al. [14] observed this same tendency in cultured PBMCs in allergic adult patients. (4) Stimulation of control PBMC’s using our tested substances produced no significant differences, thus indicating that COX-2 mRNA gene expression is not affected in non-allergic children even after stimulation with inflammatory factors [14]. Finally, our analyses also confirm O’Neill and Ford-Hutchinson’s [27] assertion that *COX-2* mRNA gene expression is normally co-expressed at detectable levels in human tissues. 

Although there is no available literature describing COX-2 secretion in PBMCs isolated from healthy and allergy children, our study confirmed COX-2 presence in the media. In addition, histamine stimulated PBMC’s registered significant statistical difference in our controls compared to the allergy group. Further, Cianchi et al. [28] reported that prostaglandin E_2_ (PGE_2_), the main product of COX-2 activity, promotes molecular mechanisms in allergic inflammation. 

Using a cell-free in vitro COX-2 screening assay, it was found that COX-2 significantly inhibited osthole and the reference compound DuP-679. DuP-679 is a member of the diaryl heterocycle group of selective COX-2 inhibitors. Furthermore, we did not observe statistically significant differences between inhibition of DuP-679 and osthole (Figure 2A). These findings indicated that there may be the same modes of action of osthole as in DuP-673 on the COX-2 inhibitory effect. This finding supports the additional pharmacological activity and confirms our earlier hypotheses about the antiallergic effect of osthole.

To the best of our knowledge, there is no previous study of interaction between histamine and COX-2 in allergy and herein we provide the first demonstration that exogenous histamine increases COX-2 PBMC protein secretion in both control and allergy groups and also increased activity of COX-2 in stimulated cells. Our results also indicate that osthole prevents histamine-induced COX-2 overexpression in the allergy group. Sousa et al.’s. [29] report stating that non-steroidal anti-inflammatory drugs inhibit COX-2 activity supports our conclusion that wider knowledge of this statistically significantly decreased COX-2 secretion in osthole-cultured cells will benefit anti-inflammatory therapy.

Results from the measured COX-2 serum concentration in both groups confirmed a significant 2-fold increased COX-2 concentration in allergic patients’ sera compared to controls.

Researchers report that IL-1β serum levels can be elevated in inflammatory and non-inflammatory conditions as different as sepsis, cancer, chronic rheumatoid disease, allergic inflammation, and non-inflammatory tissue injury, and also that PBMC IL-1β secretory changes in various conditions lead to changes in their body fluid IL-1β concentrations [14,15,16,17]. Our study produced similar results, where allergy group serum IL-1β concentration was significantly higher than controls. Some effects of IL-1β can be mimicked using cAMP-increasing agents, which activate protein kinase A (PKA) [30], and although PGE_2_ is mediated by the four EP_1_–EP_4_ receptors, only EP_2_ and EP_4_ activation increases cAMP levels [31].

The combined results suggest that defective PGE_2_-related induction of IL-1RI occurs because high EP_2_ receptor expression in the allergy group PBMCs results in increased cAMP production. Our findings show that the increased expression of IL-1RI reported in patients with allergies is closely related to their accompanying high EP_2_ receptor expression. This suggests that abnormal regulation of the autocrine loop regulating the COX pathway, which includes IL-1RI, COX-2, EP_2_, and PGE_2_, increases PGE_2_ production in allergic patients (Figure 5). Moreover, the high EP_2_ receptor expression appears to play a central role in the dysregulated autocrine loop. This is deduced from the finding that the entire mechanism normalizes on restoration of normal EP_2_ receptor function (Figure 6).

## 4. Material and Methods

### 4.1. Participants

The patients were recruited by specialists in the Regional Specialized Children’s Hospital in Olsztyn, Poland. Informed consent was obtained from all children’s parents and the study was approved by the Local Bioethics Committee (No. 19/2016; 18/5/2016).

Our control group was comprised of 28 healthy children with no history of behavioral disorders and the study group consisted of 30 children with diagnosed allergies. Children with fever, infections, and skin problems, and those taking steroids or antibiotics, were excluded from the study. The patients were allowed β2 antagonists when necessary, but no medications were taken 72 h prior to blood collection and all other medications, including anti-histamines, were excluded during the study. Patients were selected based on established criteria: diagnosed asthma, skin prick test, means of cIgE, and a-sIgE in the blood, as well as blood eosinophilia. Demographic and clinical characteristic of the study population are shown in Table 1.

### 4.2. Biological Material

A sample of 2–5 mL of serum in 5–10 mL peripheral blood was collected from each patient by medical staff at the Regional Children’s Hospital in Olsztyn. All biological material was immediately transported to the laboratory and used in analysis or stored at −80 °C. 

### 4.3. Examined Substances 

(1) Histamine (PubChem CID: 774) was obtained from Sigma-Aldrich, St. Louis, MO, USA and dissolved in RPMI-1640 (Sigma-Aldrich, St. Louis, MO, USA). The concentration of 150 ng/mL histamine was used for PBMC stimulation in an in vitro model.

(2) Osthole (PubChem CID: 10228) was obtained from Sigma-Aldrich St. Louis, MO, USA and dissolved in 96% ethyl alcohol (AbChem, Olsztyn, Poland). Herein, 300 ng/mL osthole was used.

(3) DuP-697 (PubChem CID: 3177) was obtained from Cayman Chemical Company, Ann Arbor, USA). DuP-697 was dissolved in 96% ethyl alcohol (AbChem, Olsztyn, Poland) and we also used a concentration of 300 ng/mL to compare our results with FXF and osthole.

(4) The substance mixture was prepared with a ratio of 1:2 (*v/v*).

All solutions were sterilized through a 0.22 μg/mL filter and stored at 4 °C as stock solutions for later dilution. The chemical structures of these compounds are demonstrated in Figure 7.

### 4.4. PBMC Isolation

Subject blood was collected in K_3_ETDA tubes (BD, Biosciences, San Jose, Canada) and PBMC isolation began immediately. Cell collection by Histopaque reagent (Sigma-Aldrich, St. Louis, MO, USA) was as in Kordulewska et al. [16]. Isolated cells were seeded on 24-well plates (Sarstedt, Nümbrecht, Germany) at 1 × 10^6^ per well with RPMI-1640 (Sigma-Aldrich, St. Louis, MO, USA) and supplemented with 1% heat inactivated human AB serum, 1% gentamicin, and 0.25% PHA (Sigma-Aldrich, St. Louis, MO, USA). Active reagents were added to each well after 24 h and a pure medium formed the control for each substance. Cells were then harvested after a further three days.

### 4.5. Analysis of IL-1RI, EP_2_, and COX-2 Gene Expression 

RNA was isolated from PBMCs as in Kordulewska et al. [15] with TRIzol Reagent (Sigma-Aldrich, St. Louis, MO, USA) and the conditions for all real-time PCR analysis were optimized at a 55–62 °C melting point and primer concentrations before commencing relevant experiments.

*IL-1RI*, *EP_2_*, and *COX-2* gene expression and housekeeping human β-actin gene (*ACTB*) were examined, with *ACTB* used as the reference gene to normalize differences in total RNA amounts in each sample. Oligonucleotide primers specific to each gene were designed with Primer-BLAST and PCR primers. In our manuscript we measured only gene expression and not protein expression.

### 4.6. Measurement of COX-2 Concentration

COX-2 concentrations in medium were evaluated by commercial enzyme-linked Immunosorbent Assay Kit for Prostaglandin Endoperoxide Synthase 2 (PTGS2) (Cloud-Clone Corp, Katy, TX, USA) according to the manufacturer’s instructions. Briefly, we prepared all reagents, samples, and standards according to the manual instructions of the kit. First of all, we added 100 µL standard or medium with PBMC cells to each well and incubated for 1 h at 37 °C. Then aspirated and added 100 µL prepared detection reagent A and incubated for 1 h at 37 °C. After that time, we washed three times and added 100 µL prepared detection reagent B. Then, incubated for 30 min at 37 °C, and again washed five times. We then added 90 µL substrate solution and incubated for 20 min at 37 °C and added 50 µL stop solution. The absorbance at 450 nm was read immediately using a microplate reader and the concentration of COX-2 was calculated according to the manufacturer’s instructions. 

### 4.7. Assay of COX-2 Enzymatic Activity

The in vitro inhibitory activity of histamine, FXF, osthole, and DuP-697 on purified COX-2 enzyme was determined using a colorimetric COX inhibitor screening assay kit according to the manufacturer’s instructions (Cayman Chemical Company, Michigan, USA). Briefly, a 160 μL assay buffer and 10 μL heme were added to the background wells, while a 150 μL assay buffer, 10 μL heme, and 10 μL COX-2 enzyme were added to the 100% initial activity wells. Tested substances in 10 μL at final concentrations were added to the sample wells and 10 μL of medium RPMI was added to the background wells. The plate was carefully shaken for a few seconds and incubated for five min at 25 °C. The colorimetric substrate solution (20 μL) followed by arachidonic acid (20 μL) were added to each well. The plate was again shaken carefully for a few seconds and incubated for 5 min at 25 °C. The absorbance at 590 nm was read using a microplate reader and the inhibition ratio of COX-2 enzymatic activity was calculated according to the manufacturer’s instructions.

### 4.8. Measurement of IL-1β

Commercial ELISA (Diaclone, Besancon Cedex, France) kits determined interleukin IL-1β via a quantitative sandwich immunoassay, with kits used according to manufacturer instructions. Triplicate samples were run, and the results were equalized via comparison with standard curves expressed in pg/mL. Briefly, 24 h before the measurement of IL-1β, we prepared the plate. To every well, we added 100 μL of diluted capture antibody, covered with a plastic plate, and incubated at 4 °C overnight. After that time, we washed the plate three times (400 μL). Then, we added 250 μL of blocking buffer to every well, covered with a plastic plate cover, and incubated at room temperature for 2 h. Again, we washed three times (400 μL) and prepared a standard curve (from 500 pg/mL to 15.6 pg/mL) and samples. We added 100 μL of each standard, samples, and zero to appropriate wells in triplicate. Then, we added 50 μL of diluted detection antibody into all wells, covered with a plastic plate, and incubated at room temperature for 3 h. After that time, we washed three times as before. Then, we added 100 μL of Streptavidin – HRP solution into the wells, covered with a plastic plate, incubated at room temperature for 30 min, and washed like before. We then added 100 μL of ready-to-use TMB substrate solution into all wells. Then, incubated in the dark for 30 min at room temperature. We avoided direct exposure to light by wrapping the plate in aluminium foil. We then added 100 μL of H_2_SO_4_: stop reagent into the wells. The absorbance using 450 nm as the primary wavelength and the 620 nm as the references wavelength was read using a microplate reader, and the amount of IL-1β was calculated according to the manufacturer’s instructions. The amount of IL-1β in each sample was determined by extrapolating OD values against IL-1β standard concentrations using a standard curve. 

### 4.9. Statistical Analysis

All statistical analyses were performed in triplicate using GraphPad Prism version 6.0 (GraphPad Software, Inc., San Diego, CA, USA), with results presented as mean ± SEM, and the mean values between control and allergy groups were compared using:

(1) ANOVA test (*p* < 0.05, 95% confidence interval) *for HRH-1*, *IL-1RI*, *EP_2_*, and *COX-2* gene expression and IL-1β and COX-2 level in medium after stimulation by the tested substances.

(2) Unpaired *t* test (*p* < 0.05, 95% confidence interval) with equal S.D. for COX-2 and IL-1β concentration and % COX-2 inhibition in medium with PBMC cells after stimulation by the tested substances.

(3) Kruskal–Wallis test (*p* < 0.05) for % COX-2 inhibition in RPMI medium.

## 5. Conclusions

These results and reports show that defective PGE_2_-related IL-1RI induction was due to EP_2_ receptor expression in the allergy group PBMCs, which results in increased cAMP production. Our findings show that increased expression of IL-1RI in patients with allergy is closely related to high EP_2_ receptor expression. Abnormal regulation of the autocrine loop regulating the COX pathway may increase PGE_2_ production in allergic patients. This loop includes IL-1RI, COX-2, EP_2_, and PGE_2_. High EP_2_ receptor expression therefore has a central role in the dysregulated autocrine loop. This is proven by an effective mechanism action when normal EP_2_ receptor function is restored.

## Figures and Tables

**Figure 1 ijms-20-00186-f001:**
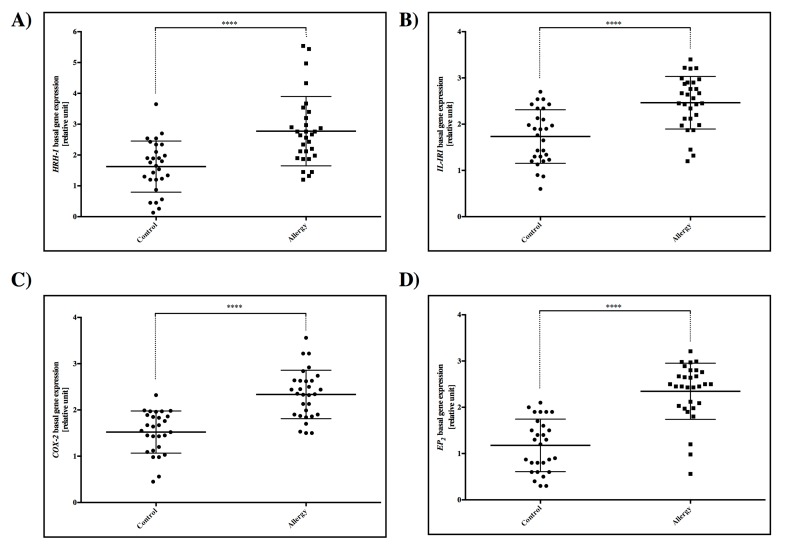
Basal gene mRNA expression of (**A**) *HRH-1* receptor, (**B**) *IL-1RI* receptor, (**C**) COX-2, and (**D**) *EP_2_* receptor in cultured PBMC from the control and allergy groups. Statistically significant differences between the control and tested sample are directly above the error bar: **** *p* < 0.0001.

**Figure 2 ijms-20-00186-f002:**
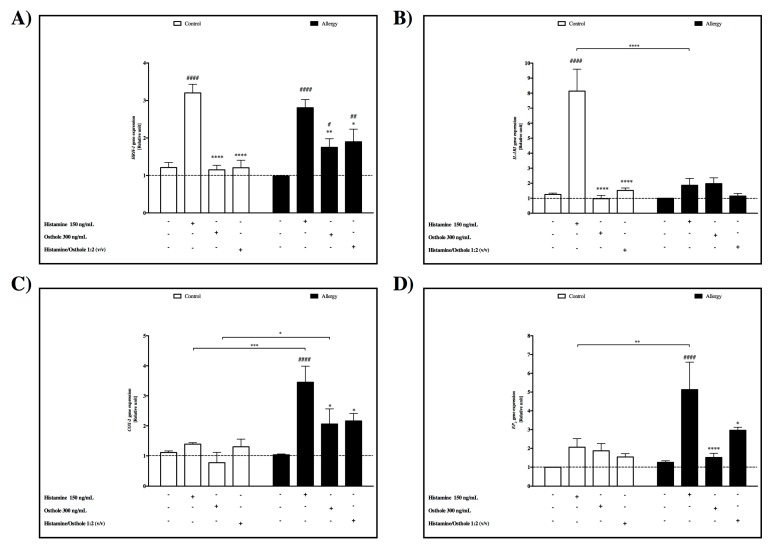
Comparison of mRNA gene expression changes in control and allergy group PBMCs under the influence of histamine, osthole, and histamine/osthole between the control and allergy group. (**A**) *HRH-1* receptor, (**B**) *IL-1RI* receptor, (**C**) COX-2, and (**D**) *EP*_2_. The control is gene expression in native cells, presented as 1. Statistically significant differences between the control and tested sample are directly above the error bar: Statistically significant differences between the control and tested sample are directly above the error bar: **** *p* < 0.0001.

**Figure 3 ijms-20-00186-f003:**
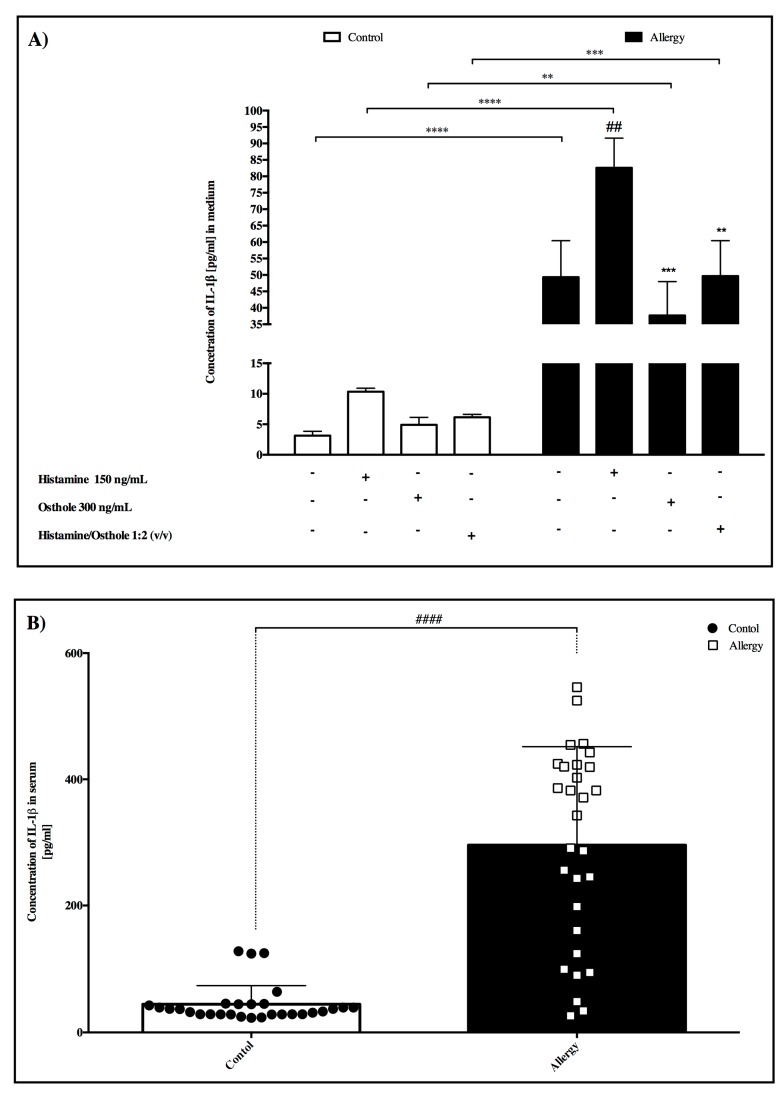
Concentration of IL-1β (**A**) in PBMC after incubation with tested substances. Here, IL-1β secretion in pure medium forms the control and tested substance secretions. (**B**) Concentration of IL-1β in serum between control and allergy group. Data are presented as the mean ± S.E.M. ## *p* < 0.01, vs. control; ** *p* < 0.01, *** *p* < 0.001, **** *p* < 0.0001 vs. treated histamine cells.

**Figure 4 ijms-20-00186-f004:**
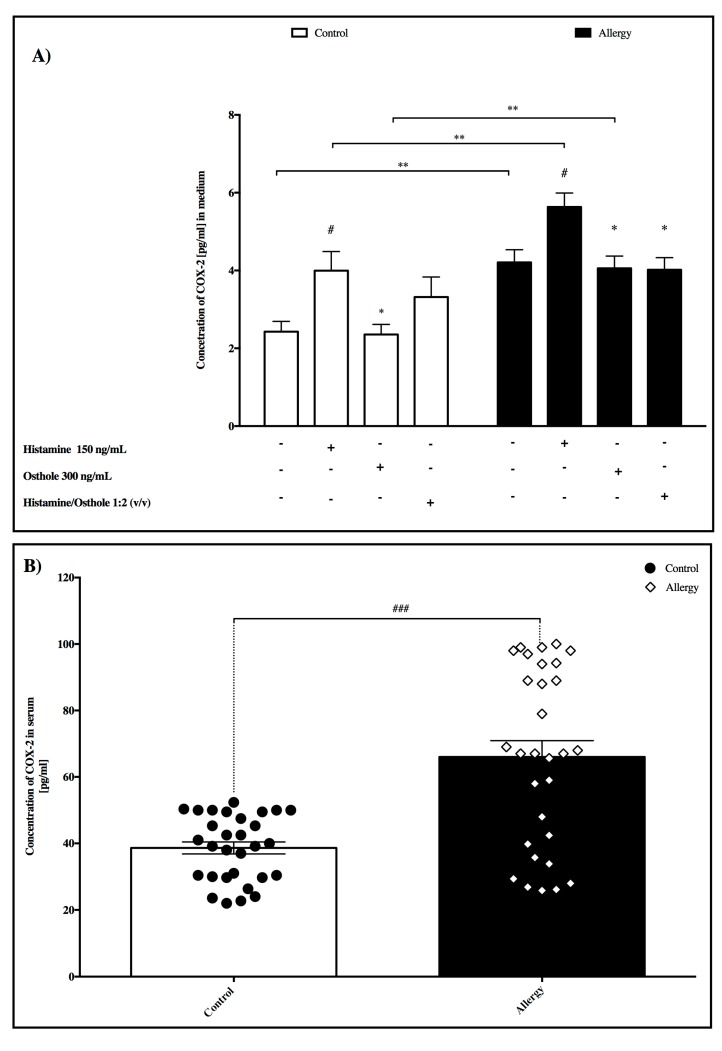
Concentration of COX-2. (**A**) Influence of tested substances on PBMC, and (**B**) in serum between control and allergy group. Here, PBMC COX-2 secretion in pure medium forms the control and tested substance secretions. Data are presented as the mean ± S.E.M. # *p* < 0.05, ### *p* < 0.001 vs. control; * *p* < 0.05, ** *p* < 0.01 vs. treated histamine cells.

**Figure 5 ijms-20-00186-f005:**
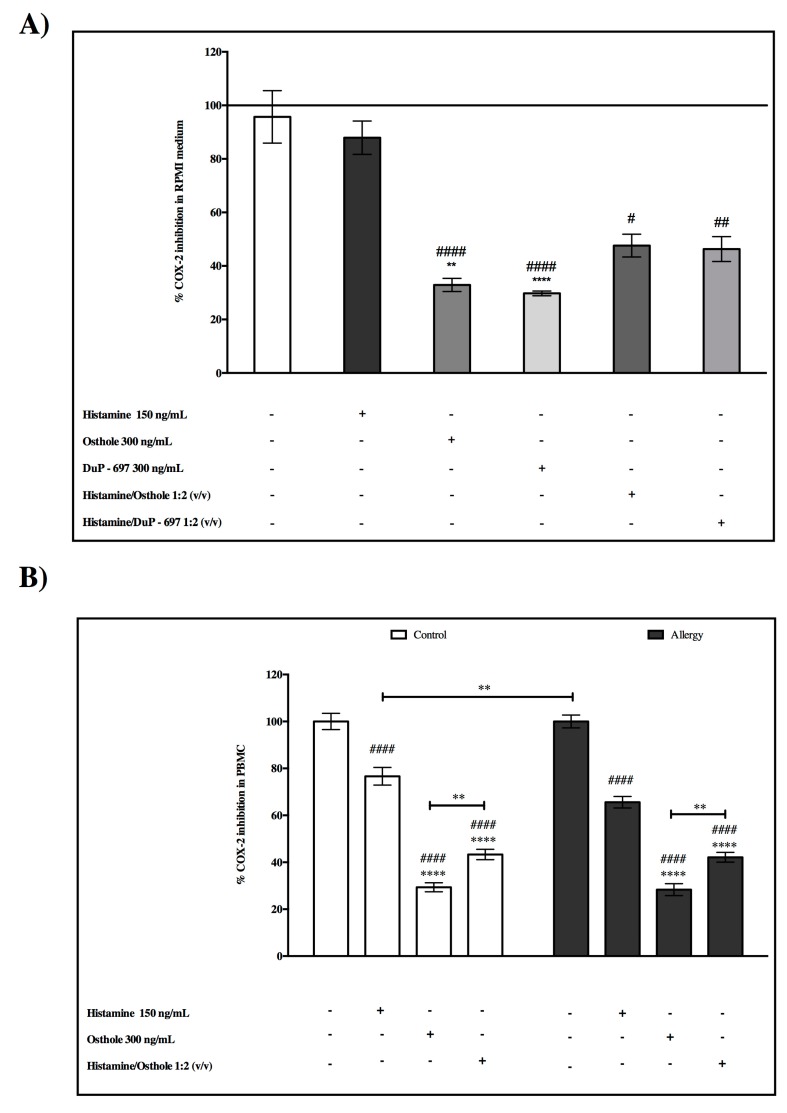
Activity of COX-2 after incubation with tested substances (**A**) in medium, and (**B**) in PBMC from the control and allergy groups. Data are presented as the mean ± S.E.M. #### *p* < 0.0001 vs. control; ** *p* < 0.01, **** *p* < 0.0001 vs. treated histamine cells.

**Figure 6 ijms-20-00186-f006:**
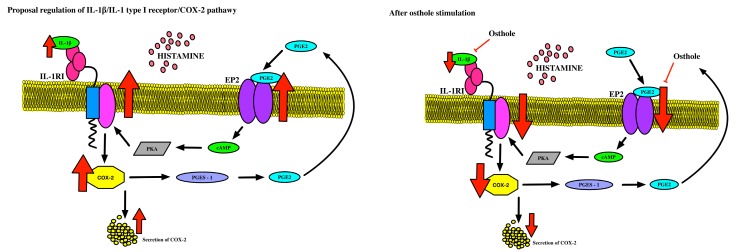
Simplified pathway of the idea of system underlying regulation of IL-1RI: (**A**) in children with diagnosed allergies, and (**B**) after osthole stimulation. IL-1β is associated with a large liking toward receptor type I and stimulated COX-2 expression, and after that, secretion. This can be allowed to alter PGE_2_ increases activity of EP_2_ receptor and activate the cAMP messenger. This likely stimulates PKA, which can repeatedly stimulate IL-1RI expression, which signifies possible positive mechanisms for this pathway. Osthole successively decreases the amount of IL-1β, decreases COX-2 expression/production, and also EP_2_ expression, which creates a possible dangerous spiral in patients with allergies (scheme based on Machado-Carvalho et al. 2017 with own modifications [32]).

**Figure 7 ijms-20-00186-f007:**
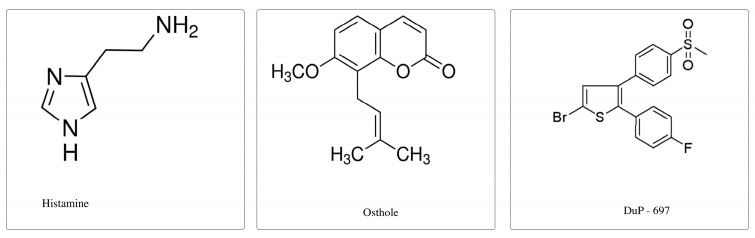
Chemical structures of histamine, osthole, and DuP-679.

**Table 1 ijms-20-00186-t001:** Demographic data and clinical characteristics of the study population. Data are presented as the mean ± S.E.M. **** *p* < 0.0001 control group vs. children with diagnosed allergies.

Characteristics	Control	Allergies
Patient’s samples, no.	28	30
Age (y), mean ± SD	7.60 ± 1.93	4 ± 1.75
Female sex, no. (%)	10 (35,7%)	8 (26,6%)
Diagnosed Allergy, no. (%)	0 (0%)	30 (100%)
Moderate/severe asthma, no. (%)	0 (0%)	15 (50%)
Skin prick test positivity, no. (%)	0 (0%)	30 (100%)
cIgE [IU/mL] mean ± SD	90.13 ± 43.93 ****	307.7 ± 130.0
a-sIgE [IU/mL] class	Negative predictive value, class 0	Positive predictive value, class 4–6
Blood eosinophilia (mean % of eosinophils in the blood smear) mean ± SD	3.668 ± 1.104 ****	23.92 ± 5.587

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
