# Peer review of "High Expression of IL-1RI and EP2 Receptors in the IL-1β/COX-2 Pathway, and a New Alternative to Non-Steroidal Drugs—Osthole in Inhibition COX-2"

_ijms, 2019, doi:10.3390/ijms20010186_

Round 1
Reviewer 1 Report
In this manuscript, the authors investigated the impact and biological activity of osthole of the expression of IL-1 type I and E-prostanoid (EP) 2 receptor, and the production and activity of COX-2 and IL-1 in the incubated medium. The results showed that osthole successively inhibited PGE2 and COX-2 mRNA expression, and reduced the secretion of protein COX-2 in signaling cellular. Changed EP2 expression in children with allergy provided higher IL-1RI induction.
I recommended accepting this paper after major revision. The following issues need to be revised.
1. The title of Figure 1 does not agree with the content of Figure 1. It just showed three chemical structure.
2. The unit ng/ml should be corrected to ng/mL.
3. Although the detailed inflammatory mechanism is clear, it is suggested to add a figure to summarize the positive impact of osthole on the inflammatory pathway based on the current results.
Author Response
DETAILED RESPONSE TO THE REVIEWERS#1:
Answer:We would like to thank to the Reviewer for taking the time reviewing this work. We improved our manuscript according to the Reviewer’s suggestions.
Comment 1: The title of Figure 1 does not agree with the content of Figure 1. It just showed three chemical structure.
Answer: We agree with the Reviewer and have corrected a text in the manuscript and a title of Figure 1.
Comment 2: The unit ng/ml should be corrected to ng/mL.
Answer:We agree with the Reviewer and have corrected this error throughout the manuscript.
Comment 3: Although the detailed inflammatory mechanism is clear, it is suggested to add a figure to summarize the positive impact of osthole on the inflammatory pathway based on the current results.
Answer: We agree with the Reviewer and have prepared the Figure 7 and Graphical Abstract. Figure 7 simplified pathway of system underlying regulation of IL-1RI idea: A) in children with diagnosed allergies, B) after osthole stimulation. IL-1β is associate with large liking to receptor type I and stimulated COX-2 expresion and after that secretion. This can be allowed to alter PGE2increases activity of EP2receptor and activate the cAMP messenger. This likely stimulate PKA which can repeatedly stimulate IL-1RI expression, by that signify a possible positives mechanism for this pathway. Osthole successively decrease amount of IL-1β, decreases COX-2 expression/production and also EP2expression which possible change dangerous circle in patients with allergy (scheme based on Machado-Carvalho et al. 2017 with own modifications).
Reviewer 2 Report
Authors have hypothesized that “ osthole has potential in allergy treatment in inhibition in 29 COX-2 pathway”. They performed a clinical study by collecting blood samples and assessing in vitro effects of osthole on PMBC.
First of all, a bibliographic search shows that the effect of osthole in reducing COX-2 expression was already found, (as reviewed by Zhang 2015,and as shown more recently by Singh 2018). Moreover, it has been already shown that Osthole treatment ameliorates Th2-mediated allergic asthma. Thus, the objectives are not novel.
In order to assess the effects of osthole on COX-2 activity, I suggest to measure PGE2 levels, as an index of COX-2 activity.
In vitro experiments should be performed in order to evaluate the inhibitory effects of compounds on LPS-stimulated monocytes or in human whole blood stimulated with LPS, which represent assay to test their effects on catalytic activity of COX-2.
Western blot analysis should be performed and not only the evaluation of mRNA of COX-2, EP2 etc.
Authors conclude that “These results and reports show that defective PGE2-related IL-1RI induction is due to EP2 2 receptor expression in allergy group PBMC’s which results in increased cAMP production.” They should show the evaluation of cAMP.
Moreover they wrote that their findings show that increased expression of IL-1RI in patients with allergy is closely related to high EP2 receptor expression. Thus, they should specify that they measure gene expression and not protein expression
Figures are very little and they are not easily comprehensive.
Figure 1 is wrong, in fact it reports some chemical structures differently as described by the legends.
A deep revision from an expert of English language is required.
Author Response
DETAILED RESPONSE TO THE REVIEWERS#2:
Answer:We would like to thank to the Reviewer for taking the time reviewing this work. We improved our manuscript according to the Reviewer’s suggestions.
Comment 1: In order to assess the effects of osthole on COX-2 activity, I suggest to measures PGE2 levels, as an index of COX-2 activity.
Answer: This is very interesting issue. According to Reviewer’s suggestion the results of COX-2 concentration in medium were added in Figure 5. These are preliminary studies, but next research will include PGE2 levels as a great index of COX-2 activity.
Comment 2: In vitro experiments should be performed in order to evaluate the inhibitory effects of compounds on LPS-stimulated monocytes or in human whole blood stimulated with LPS, which represent assay to test their effects on catalytic activity of COX-2.
Answer: Thanks also for that comment. In our experiment we focused on histamine and allergic reactions in allergic patients – that is way we stimulated only histamine our PBMC. According to Reviewer’s suggestion, during planning our next research steps we will include examination of compounds effects on LPS-stimulated PBMC’s.
Comment 3: Western blot analysis should be performed and not only the evaluation of mRNA of COX-2, EP2 etc.
Answer: The suggestion is very important for us, and in the next manuscript Western blot analysis will be included. Now we do not possess enough amount of biological material to prepare Western blot analysis. In this paper we included results from the ELISA test, where we determined IL-Iβ protein concentration (Figure 4 A,B) and COX-2 (Figure 5 A, B) by ELISA, whereas mRNA expression was used for HRH-1 analysis (Fig. 2A), IL-1RI (Fig. 2B), COX-2 (Fig. 2c) and EP2(Fig. 2D).
Comment 4: Authors conclude that “These results and reports show that defective PGE2-related IL-1RI induction is due to EP2 2 receptor expression in allergy group PBMC’s which results in increased cAMP production.” They should show the evaluation of cAMP.
Answer: We made this conclusion after obtaining the results and nowadays literature data. We agree with Reviewer’s comments, that we suppose to show the evaluation of cAMP.The suggestion is very important for us, and in the next manuscript evaluation of cAMP will be included.
Comment 5: Moreover, they wrote that their findings show that increased expression of IL-1RI in patients with allergy is closely related to high EP2 receptor expression. Thus, they should specify that they measure gene expression and not protein expression
Answer:Gene expression analysis was tested for HRH-1 (Fig. 2A), IL-1RI (Fig. 2B), COX-2 (Fig. 2c) and EP2(Fig. 2D), but protein concentration (using ELISA test) was measured for IL-Iβ protein concentration (Figure 4 a,b) and COX-2 (Figure 5 ab).
Comment 6: Figures are very little, and they are not easily comprehensive.
Answer: We prepared figures in better resolution, but figures were reduced by the editorial program during submission of manuscript. The final version of figures will be consulted with editorial office. We can send to You figures prepared in tiff in high resolution.
Comment 7: Figure 1 is wrong, in fact it reports some chemical structures differently as described by the legends.
Comment 8: We agree with the Reviewer and have corrected a text in the manuscript and a title of Figure 1.
Comment 9:A deep revision from an expert of English language is required.
Answer: We made by the expert of English language grammar and text, corrections according to Reviewer’s comment.
Round 2
Reviewer 1 Report
After carefully revising and adding suitable figures, I recommend accepting it in present form.
Reviewer 2 Report
Authors have hypothesized that “ osthole has potential in allergy treatment in inhibition in 29 COX-2 pathway”. They performed a clinical study by collecting blood samples and assessing in vitro effects of osthole on PMBC.
First of all, a bibliographic search shows that the effect of osthole in reducing COX-2 expression was already found, (as reviewed by Zhang 2015,and as shown more recently by Singh 2018). These references should be added.
Moreover, it has been shown that Osthole treatment ameliorates Th2-mediated allergic asthma. Thus, the objectives are not novel.
In order to assess the effects of osthole on COX-2 activity, I suggest to measure PGE2 levels, as an index of COX-2 activity. In my opinion, the paper needs this evaluation.
In vitro experiments should be performed in order to evaluate the inhibitory effects of compounds on LPS-stimulated monocytes or in human whole blood stimulated with LPS, which represent assay to test their effects on catalytic activity of COX-2. In my opinion, the paper needs this evaluation.
Other comments:
In the legend of figure 2, specify that it is gene expression
Why did you use histamine ate the concentration of 150 ng/ml?
The legend of figure 4 is not clear to understand.
At page 11, please clarify that you are evaluating COX-2 activity.
Why did you use osthole at 300ng/ml? Also why did you use DuP at 300ng/ml?
Legend of figure 6 is not clear: please clarify.
Author Response
DETAILED RESPONSE TO THE REVIEWERS#2:
Answer: We would like to thank to the Reviewer for taking the time reviewing this work. We improved our manuscript according to the Reviewer’s suggestions.
Comment 1: First of all, a bibliographic search shows that the effect of osthole in reducing COX-2 expression was already found, (as reviewed by Zhang 2015, and as shown more recently by Singh 2018). These references should be added.
Answer: We agree with the Reviewer and have added the references in the manuscript.
Singh, G., Bhatti, R., Mannan, R., Singh, D., Kesavan, A., & Singh, P. (2018). Osthole ameliorates neurogenic and inflammatory hyperalgesia by modulation of iNOS, COX-2, and inflammatory cytokines in mice. Inflammopharmacology, 1-12.
Comment 2: In order to assess the effects of osthole on COX-2 activity, I suggest to measure PGE2 levels, as an index of COX-2 activity. In my opinion, the paper needs this evaluation.
Answer: Thanks also for that comment. This is very interesting issue. According to Reviewer’s suggestion the results of COX-2 concentration in medium were added in Figure 5. These are preliminary studies, but next research will include PGE2 levels as a great index of COX-2 activity. Now we do not possess enough amount of biological material to prepare this analysis.
Comment 3: In vitro experiments should be performed in order to evaluate the inhibitory effects of compounds on LPS-stimulated monocytes or in human whole blood stimulated with LPS, which represent assay to test their effects on catalytic activity of COX-2. In my opinion, the paper needs this evaluation.
Answer: In our experiment we focused on histamine and allergic reactions in allergic patients – that is way we stimulated only histamine our PBMC. According to Reviewer’s suggestion, during planning our next research steps we will include examination of compounds effects on LPS-stimulated PBMC’s. Now we do not possess enough amount of biological material to once again prepare this kind analysis
Comment 4: In the legend of figure 2, specify that it is gene expression.
Answer: We agree with Reviewer’s comments, and we specify legend of the figure 2.
Comment 5: Why did you use histamine ate the concentration of 150 ng/ml?
Answer: Previously, we published article [Kordulewska, N. K., Kostyra, E., Matysiewicz, M., Cieślińska, A., & Jarmołowska, B. (2015). Impact of fexofenadine, osthole and histamine on peripheral blood mononuclear cell proliferation and cytokine secretion. European journal of pharmacology, 761, 254-261], where we describe cytotoxicity of tested substances by BrdU test. We based on the published results and that is why we use this concentration.
Comment 6: The legend of figure 4 is not clear to understand.
Answer: We agree with the Reviewer and have changed the figure legend (No. 4) in the manuscript
Comment 7: At page 11, please clarify that you are evaluating COX-2 activity.
Answer: We agree with the Reviewer and clarify that we are evaluating COX-2 activity.
Comment 8a: Why did you use osthole at 300ng/ml?
Answer: Previously, we published article [Kordulewska, N. K., Kostyra, E., Matysiewicz, M., Cieślińska, A., & Jarmołowska, B. (2015). Impact of fexofenadine, osthole and histamine on peripheral blood mononuclear cell proliferation and cytokine secretion. European journal of pharmacology, 761, 254-261], where we describe cytotoxicity of tested substances by BrdU test. We based on the published results and that is why we use this concentration.
Comment 8b: Also why did you use DuP at 300ng/ml?
Answer: Because we wanted to check the same concentrations and compare the results.
Comment 9: Legend of figure 6 is not clear: please clarify.
Answer: We agree with Reviewer’s comments, and we claryfy legend of the figure 6.